# Kinetics and Thermodynamics of Thermal Degradation of Different Starches and Estimation the OH Group and H_2_O Content on the Surface by TG/DTG-DTA

**DOI:** 10.3390/polym12020357

**Published:** 2020-02-06

**Authors:** Marita Pigłowska, Beata Kurc, Łukasz Rymaniak, Piotr Lijewski, Paweł Fuć

**Affiliations:** 1Faculty of Chemical Technology, Poznan University of Technology, Berdychowo 4, PL-60965 Poznan, Poland; marita.piglowska@student.put.poznan.pl; 2Institute of Chemistry and Electrochemistry, Faculty of Chemical Technology, Poznan University of Technology, Berdychowo 4, PL-60965 Poznan, Poland; 3Institute of Combustion Engines and Transport, Faculty of Transport Engineering, Poznan University of Technology, Piotrowo3, PL-60965 Poznan, Poland; lukasz.rymaniak@put.poznan.pl (Ł.R.); piotr.lijewski@put.poznan.pl (P.L.); pawel.fuc@put.poznan.pl (P.F.)

**Keywords:** starch, TGA, hydroxyl groups

## Abstract

The main aim of this study is to estimate the kinetic and thermodynamic parameters of thermal decomposition of starches by the Coats–Redfern method. This procedure is a commonly used thermogravimetric analysis/difference thermal gravimetry/differental thermal analysis (TG/DTG-DTA) kinetic method for single rate form. The study also shows a proposed method for reactive hydroxyl groups content on the starch surface determination, and values were in range of 960.21–1078.76 mg OH per 1 g of starch. Thermal processing revealed the thermophysical properties of biomass for the kinetics of decomposition estimation. Activation energies reached the values in range of approximately 66.5–167 kJ·mol^−1^. This research also enables the determination of the temperature conditions required for becoming the desired form of material. Therefore, it is necessary to achieve the requested compact porous structure in an activation process, because in the native state, the polymer exhibits limited applications as a result of thermal decomposition, low shear stress, retrogradation, and syneresis, hence the low solubility in organic solvents. Thermodynamic parameters and reactive hydroxyl groups in this article review are innovative and have not yet been found in the literature.

## 1. Introduction

Currently, there is an enormous demand in the rapidly developing industrial field for alternative energy sources and modifications connected to applying the biodegradable materials consistent with green chemistry, because of the increasing energy crisis and changes in our climate. Thus, innovations are also involved with applications with the most natural energy transformations possible; that are environmentally friendly. Mostly, the adjustments are associated with global warming remission due to rising greenhouse gas CO_2_ concentration in our atmosphere [1,2,3]. The global humanity seeks to decrease the undesirable impact of energy transformations on human health and the environment. Starch is a natural green polymer—containing about 20–30% of long-chained amylose, and which is a complex mixture of weakly branched and strictly linear polysaccharides—that may build the semi-crystalline region and circa 70–80% of short-chained, highly branched amylopectin. Depending on the starch botanical source, the second polymer could be more responsible for the crystalline or amorphous region [4]. The lower the starch’s crystallinity, the greater its amylose content [5]. The amylopectin is responsible for the crystalline and amorphous regions as the branching points are clustered in this component of starch.

According to the literature [6], within starch granules, amylose molecules are not present as bundles in the amorphous regions. They are rather interspersed among amylopectin. The amount of those structures in the starch molecule depends of its botanical origin. Starch granules occur in a semi-crystalline densely packed form. According to the literature [7], the crystallinity reaches values in the range of 15–45%, and a density of 1.5 g·cm^−3^. Both compounds have α-O-1,4-glucosidic bands, and the structure of amylopectin additionally contains α-O-1,6-glucosidic bands. Approximately 95% of biopolymer exhibits α(1,4) linkages and only 5% α(1,6) bands [8]. Starch could be easily found, as naturally occurring energy storage, rice, seeds, wheat, peas, potatoes, corn, bananas, and topioca. Starch consists of glucose units (C_6_H_10_O_5_)_n_, where n is a value between 300 and 1000 [1]. This biopolymer is well known for its renewability, low cost, and biodegradability [2]. There are three main types of starch, categorized according to diffraction patterns under X-ray: A, B, and C. Type A is starches from cereals; type B is starches from tubers, fruits, and stems; and type C is starches from legumes and roots [9]. Types A and B are involved in the structure parallel double helices and type C is the mix of both of them in a single or distinct granule [10]. Starch shows properties characteristic of aldehydes, alcohols, and ethers. The ability to reduce is slight because only the initial glucosidic unit, one for the whole chain, has free anomeric carbon. Depending on the conditions of modifications of starches, various transformations could have been observed: retrogradation, gelatinization, glass transition, and dextrination. Dextrination leads to hydrolysis, transglucosidation, or repolymerization [11]. The transglucosidation process leads to the intermolecular bond formation, where different glucosidic linkages are created. Thermal modification, that is, pyrolysis to dextrins, destroys granules’ structure. Gelatinization and subsequent drying allow the obtaining of the polysaccharide, which swells and disperses in cold water [12]. The pyrolysis process of starch in an inert atmosphere consists of a few steps: water evaporation, when the water amount in starch depends on the adsorbed; hydroxyl groups’ degradation and thermal decomposition of organic compounds; and carbonization. The process above 600 °C could, but should not inhere in an inert gas atmosphere, and leads finally to ash. The mineral ash could be obtained only in an oxygen atmosphere. According to the work of [12], at about 180 °C, the diffraction patterns are less sharp, and at 210–220 °C, the amorphous region is produced and the birefringence of molecules is destroyed. To better understand the methods for measuring the biodegradability of starch, Figure 1 was applied. It shows that this parameter could be obtained by respirometric, morphological, spectroscopic, gravimetric, chromatographic, microbiologic, and physical and morphological techniques. In this study, the method of thermal analysis, as well as part of the physical and morphological methods, were applied because of many valuable applications, for example, in kinetics, thermal, and surface properties as well as thermodynamics, which are justified for this work.

## 2. Materials and Methods

### 2.1. Materials

All starches were purchased from Sigma-Aldrich (Oakville, ON, Canada). In research, rice starch, wheat starch, corn starch, and potato starch were applied (acc. To Zulkowsky, treated with glicerol at 190 °C). Glicerol is the 1,2,3-Propanetriol.

### 2.2. Thermogravimetric Analysis/Difference Thermal Gravimetry/Differental Thermal Analysis (TG/DTG-DTA) Experimental

The main aim was to define changes in the structure of starches from different botanical origins. It was possible to also obtain the amount of hydroxyl groups of starch molecules using TG/DTG-DTA (thermogravimetric analysis/difference thermal gravimetry/differental thermal analysis) data. To carry out the analysis, a Perkin Elmer thermogravimetric analyzer (TGA) 8000 (Krakow, Poland) was used. TG/DTG-DTA techniques were carried out in a simultaneous TG/DTG-DTA instrument. In the study, an alumina crucible without any lid was used. The TG/DTG-DTA analysis enables the accurate assessment of thermal stability, and thus the lifetime of the material, using the range of temperatures of 29.6–1062.1 °C, while the heating rate ß was 10 K·min^−1^. The gas flow rate was 20 mL·min^−1^. The process was conducted in the inert gas atmosphere (nitrogen). To conduct the analysis, amounts of 11.8, 12.3, 13.4, and 15.1 mg of starch from rice, potato, wheat, and corn, respectively, were used, and then the changes in the mass of the heat-treated material were obtained. It should be assumed that the results of the thermal analysis for these selected samples are always repeatable and the maximum number of samples depends on the instrument’s requirements. The result of analysis was shown using appropriate graphs, considering the mass loss in every 10 K per minute in a precisely chosen temperature range. The data graph, meanwhile, describes the physical and chemical transformations, such as crystallization, melting, and sublimation, and thus the phase transitions.

### 2.3. Methods Adopted to Obtain the Kinetics

To determine the kinetics of starch decomposition, the TG/DTG-DTA data were used. The fundamental equation, used in all TG/DTG-DTA calculations, is described as follows:(1)$$\frac{d\alpha }{dt}=k\cdot f\left(\alpha \right)$$

The conversion rate α is given by the following equation:(2)$$\alpha =\frac{m_{i}-m_{t}}{m_{i}-m_{f}}$$
where *m_i_* is the initial mass; *m_t_* is the mass in time *t*, constant in given temperature: K; and *m_f_* is the final mass.

The generally occurring Arrhenius equation, and thus the rate constant *k*, has the following formula:(3)$$k=\;A\exp\left(-\frac{E_{a}}{RT}\right)$$

The heating rate is given by the following formula:(4)$$\beta =\frac{dT}{dt\;}$$

To obtain the reaction rate constant, the modified Arrhenius equation, and thus the combination of Equations (1), (3), and (4), is recommended:(5)$$k\left(t\right)=\;\frac{A}{\beta }\exp\left(-\frac{E_{a}}{RT}\right)f\left(\alpha \right)$$
where *k*(*t*) is the reaction rate constant—min^−1^, *f*(*α*) is a function describing the transition (reaction model), *E_a_* is the activation energy—J·mol^−1^·K^−1^, *A* is the pre-exponential factor—min^−1^, *T* is the temperature of heating—K, and *R* is the gas constant reaching the value of 8.314 J·mol^−1^·K^−1^.

Assuming the non-isothermal conditions of pyrolysis, for the data analysis, the form of function with infinite limits *g*(*α*), given by the equation below, was used [14].
(6)$$g\left(\alpha \right)=\int \frac{d\alpha }{f\left(\alpha \right)}=\frac{A}{\beta }\int \exp\left(-\frac{E_{a}}{RT}\right)f\left(\alpha \right)$$
where the *g*(*α*) and *f*(*α*) functions for suitable mechanisms are presented in Table 1.

To designate the Gibbs free enthalpy of decomposition, the general Eyring–Polanyi equation was used.
(7)$$k=\;\frac{k_{b}\cdot T}{h}\exp\left(-\frac{\Delta G}{RT}\right)$$
where *k_b_* is the Boltzmann constant, equal to approximately 1.3806 × 10^−23^ J·K^−1^; and *h* is determined as the Planck constant, equal to circa 6.6261 × 10^−34^ J·s.

Further, to estimate enthalpy, the combination of Equations (3) and (7) was used, given by Formula (8).
(8)E=ΔH+RT

Finally, to determine entropy of degradation, the Gibbs Function (9) is used.
(9)ΔG=ΔH−TΔS

Furthermore, it was possible to calculate the equilibrium constant *K*, using the isotherm of van’t Hoff, described as follows:(10)ΔG=−RTlnK

For the graph, the best fits, as well as the slope and intercept values, are used to determine *E_a_* and *A* parameters from the equation (Coats–Redfern (CR)):(11)$$\ln\left(\frac{g\left(\alpha \right)}{T^{2}}\right)=\ln(\frac{AR}{\beta E_{a}}(1-\frac{2RT_{max}}{E_{a}}))-\;\frac{E_{a}}{RT}$$

Algebraic expression and plot recommended for the adopted kinetic model [15] with plots:(12)$$\ln\left(\frac{g\left(\alpha \right)}{T^{2}}\right)=f\left(\frac{1}{T}\right)$$

## 3. Results

### 3.1. TG/DTG-DTA

To determine the kinetics of starch, the TGA/DTG-DTA graphs were used.

The DTA and DTG data allowed to distinguish the dehydratation and dehydrogenation processes, and thus the first and second loss of mass during the process adopted. The TG analysis (Figure 2a) presents four stages of starch decomposition. It is clearly seen that the thermographs of the three starches (from wheat, rice, and corn) are really similar, so the wheat and rice starch are almost identical and have similar thermal properties. The peak maxima in the DTG (Figure 2b) curve represents the maximum rate of mass loss. The TG graph for potato starch may look different, because of differences in the structure caused by different amounts of polymer compoundsand thus the crystalline and amorphous region as well as the granule size. A degradation of this botanical origin is less severe than others, but begins earlier. Thus, the retrogradation and dextrination, according to the literature, could be more intense and the result is less mass loss after the process. According to the literature data, starches exhibit the largest thermal transition [16]. Figure 3 shows the proposed pathways during the thermal treatment of starch, and it should be noticed that there is the possibility of many more products being obtained. Additionally, the pathway, where the cyclic products are obtained, is more likely for polysaccharides.

### 3.2. Kinetics of Thermal Decomposition

To obtain the kinetics of starch degradation, graphs (ln(*g*(*α*)*T**^−^*^2^)) versus (*T*^−1^) for each *g*(*α*) were created—Equation (12). Next, to determine the parameters, the best fit (slope and intercept) using the most equal to 1 R^2^ parameter based on Table 2 was chosen. According to the literature, the process of starch degradation is connected to models A_2_, A_3_, A_4_, D_1_, and F_1_. For every starch, the best correlation (Table 2) was obtained for model F_1_, which means the first-order reaction, and so that model was used for the next calculations. Using the slope and intercept, it was possible to obtain the activation energy and *A* parameter, using the CR model—Equation (11). The results enabled obtaining other thermodynamic parameters (free energy, free enthalpy, and entropy) for the reaction. To obtain those parameters, Equations (8)–(10) were used. Those parameters include the second step of the TG graph, which means when the hydroxyl groups are degraded (±280–360 °C). A_2_, A_3_, A_4_, D_1_, and F_1_ in Figure 4a–d are adopted as well as the featured mechanisms from Table 1.

Table 3 presents the kinetic and thermodynamic parameters of starch degradation. The value of Gibbs free enthalpy is far higher than zero. Thus, the reaction can proceed to the left, so the process is absolutely forced. Figure 5a–d show directly proportional, quite linear dependence on the temperature Gibbs function. This result confirms the high dependence on the emperature of thermodynamic parameters. According to Table 3, the positive value of enthalpy indicates the endothermic transformation, meaning with energy absorption. The negative value of entropy shows the undesirable nature of the transformation, that the entropy of the reaction decreases while the degrees of freedom increase. The positive value of enthalpy shows the dependence, that the rise of temperature causes shifting of the balance towards product creation (according to the Le Chatelier–Braun rule). The small value, lower than zero, of K means that the reaction is irreversible. According to the thermographs, the second mass loss is the most important in this research.

This decrease concerns the chemical dehydration and thermal deposition of starch molecules. The degradation reactions begin at the temperature of circa 280 °C, when the thermal condensation between hydroxyl groups of starch chains starts the formation of ether fragments, and the release of water molecules is obtained. When dehydratation is located in the neighborhood of itself, hydroxyl groups in the glycosidic ring caus the formation of the C=C bond or degradation of the glycosidic ring. All of aldehyde groups are formed at the same time as terminal groups, while the monosaccharide ring is damaged. Aromatic rings such as substituted benzene and furan structures with groups such as –CH_2_– or –CH_2_–O–CH_2_– as main binders between the aromatic rings are made, while the temperature of degradation increases [14].

It is notable that, above the temperature range of 550–600 °C, where the amorphous region has been built, the mobility of ions should be increased, so the increased thermodiffusion and diffusion processes could be reached. The really slight mass loss of the third loss could be also connected to really strong C–C covalent bonds of amorphous carbon. According to Table 3, some differences in the activation energy of potato starch degradation may be connected to modifications with glycerol made by the producent. So the potato starch, according to the TG/DTG-DTA graph, had degraded before other starches with similar trends. This may be caused by less moisture or a lower carbon content in the structure. Furthermore, this modification could increase the ability of the retrogradation process during the heating, so a channel of the crystalline region could become more hydrophobic than the channel of the amorphous area. This non-water channel may decrease the ability to hold water molecules together and separate them [18]. The common phenomena during the dry heating is the dextrination process, where dextrins are obtained and the material has a brown color. It has been noticed, according to Table 3, that the energy to begin the decomposition decreases in the turn of corn > rice > wheat > potato, which is in agreement with the TG/DTG-DTA graphs. Thus, the least energy-consuming is potato starch, and the most energy-consuming is corn starch. The researchers indicated that higher amylopectin starch has more stability. The reason for the higher activation energy of corn starch could be the higher amylopectin content and more stable microstructures, which need more energy to break the starch molecular chains [14]. In addition, Figure 4d does not have the linear trend that Figure 4a–c has, which may confirm a rather multi-step degradation reaction for corn starch. The value of the reaction rate constant is the highest for wheat starch and lowest for corn starch. It should be noticed that the heating rate constant ß has a significant impact on kinetics, and its higher value could change kinetics values. Using the TG/DTG-DTA-FTIR analysis, according to the literature data, the following releasing gases after pyrolysis could be seen u.a: H_2_O, C=O, CH_4_, C_2_H_2_, and C_2_H_4_O_2_ [14]. Pyrolysis of starches in this step is connected to the release of water, carbon dioxide, carbon monoxide, acetaldehyde, furan, and 2-methyl furan [17]. The kinetics of starch degradation, and thus the activation energy and pre-exponential factor, were determined in the literature [15,17,19]. The main monomer of starch is the glucose, so the pyrolysis of this monosaccharide has a significant implication on the pyrolysis of many polysaccharides in order to better understand the kinetics of their degradation. The values of the activation parameters are much lower than for starch, but this confirms that polymer consists of many glucose units in the structure, and so to activate the process of degradation, higher energy should be used [20]. It should be also noticed that the determination of the thermodynamic parameters for the second mass loss of the degradation of starches could not be found in the literature. Only gelatinization, transition, and melting enthalpies determined by DSC (differential scanning calorimetry) analysis could be seen, and thus the changes of the phases or state [21].

### 3.3. Determination of Hydroxyl Groups of Starches Using Second Mass Loss From TGA

To quantify the hydroxyl group content on the starch molecule area, Equation (13) is used.
(13)$$n_{OH}{\left(C_{6}H_{10}O_{5}\right)}_{n}=2n_{H_{2}O}=\frac{2\left(WL\left(T_{0}\right)-WL\left(T_{final}\right)\right)}{100M_{H_{2}O}}$$
where $WL\left(T_{0}\right)-WL\left(T_{final}\right)$ is the mass loss in % in the region of *T_o_*–*T_final_*; and $M_{H_{2}O}$ is the molar mass of water, reaching the value of 18.015 g·mol^−1^. For calculations, the hydroxyl group OH molar mass (M_OH_) has the value of 17.008 g·mol^−1^.

The total hydroxyl group content was calculated from the entire second mass loss, so after all physisorbed water was removed from the structure, and all values are shown in Table 4.

It was observed that the fewer the hydroxyl groups that stayed on the structure after the second step, the more stable the starch was, which was confirmed by the literature [23]. According to Table 4, all starches contain many hydroxyl groups in range of 960.21 up to 1078.76 mg·g^−1^ on the starch surface, which confirms their belonging to carbohydrates, which contain circa 60–70% of water, including the hydroxyl groups, and 30–40% of black carbon. Those values depend on the starch origin. Comparing starch to disaccharides such as saccharose, starch contains more water in the range of 6.23–9.83% and, per 1 g of starch, those values are in the range of 62.30 up to 98.28 mg; the values are presented in Table 4. For the water physisorbed for the saccharose, the horizontal straight line in first mass loss was obtained, because of the lower water content in the structure, practically equal to approximately zero (<0.1%). The moisture in this case is eliminated up to 200 °C [24]. For starches, all physisorbed water eliminated in range of 202–270 °C and the first mass loss is not that horizontal, which confirms TG/DTG-DTA (Figure 2a). Because of the high amount of water and hydroxyl groups on the starch surface, according to Table 4, the pyrolysis process leads to the degradation of material without high environmental pollutions, and so the starch, as desired, is biodegradable. It should be remarked that the biodegradation grad depends on the physical and chemical parameters, such as pH, moisture, temperature, air content, and nutrients [13]. The biodegradation of starch could be examined through some different methods from the work of [13] and Figure 1. The reactive hydroxyl group content on starch surface was not found in the literature, so it could be an innovative method to determine. In the work of [25,26], the OH content in cellulose was determined as a value of 1037 mg OH per gram of polymer. Because of the similar structures of those polymers, in this research, the values obtained may be very likely. In addition, the lower amount of hydroxyl groups of potato starch could be connected with the 1,2,3-trihydroxypropanol modification, where glycerol may react with hydroxyl groups from the starch molecule and decrease the reactive OH amount on the starch surface. It should have been seen that the physisorbed water content could not be determined precisely by thermal gravimetric analysis, because before the process had begun, the part of water was instantly released with the running nitrogen gas flow [22].

## 4. Conclusions

This study made it clear that TGA/DTG-DTA was suitably used to estimate the kinetics and thermodynamic parameters of starch degradation, and thus the amount of reactive hydroxyl groups on the surface of molecules. Really large mass loss during the process depends on the structure of polymer, and thus the chain length and functional groups transitions. Thus, the higher activation energy for corn starch could be the result of the possible higher molecular weight and thus more α-1,6 linkages. It may be attributed to the more advanced amylopectin content. Additionally, DTA shows endothermic character of the reaction, which the calculated kinetics and thermodynamics confirm. The obtained results of the kinetics of thermal degradation are mathematically and physically not synonymous with typical first-order kinetic reactions, which allows obtaining the half-life of materials from classic equations, and so kinetic formulas for the research were modified.

## Figures and Tables

**Figure 1 polymers-12-00357-f001:**
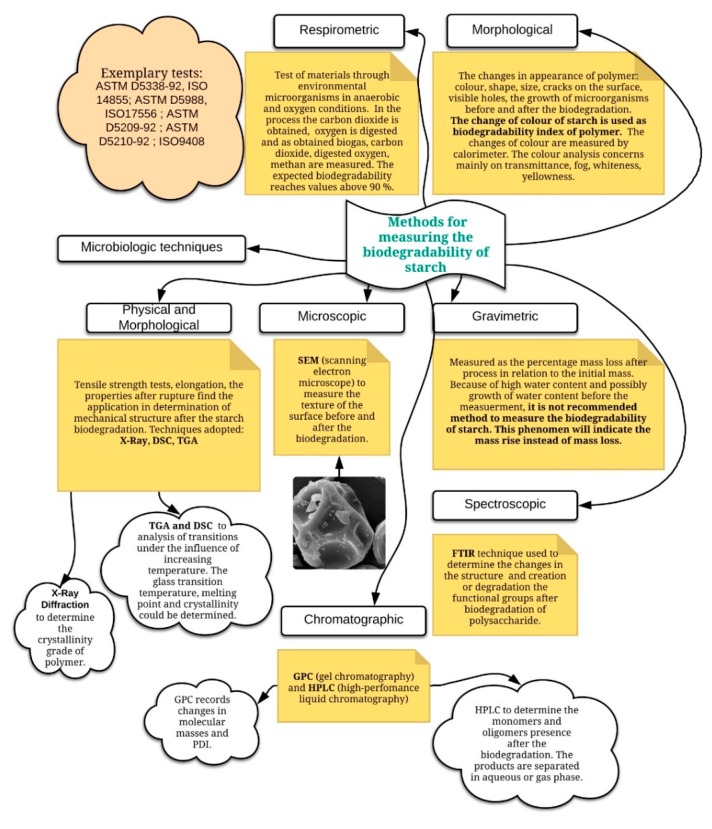
Methods for measuring the biodegradability of starch [13].

**Figure 2 polymers-12-00357-f002:**
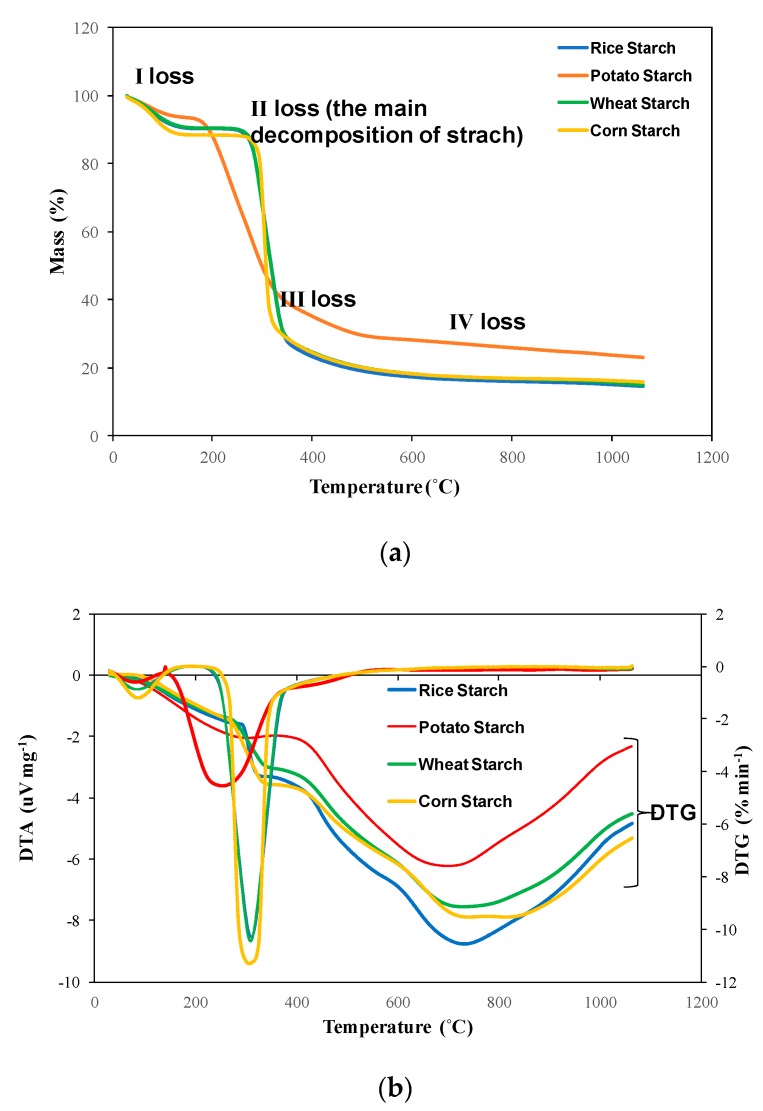
Thermogravimetric analysis (TGA) comparing native starches (**a**), and differental thermal analysis (DTA) and difference thermal gravimetry (DTG) comparing native starches (**b**).

**Figure 3 polymers-12-00357-f003:**
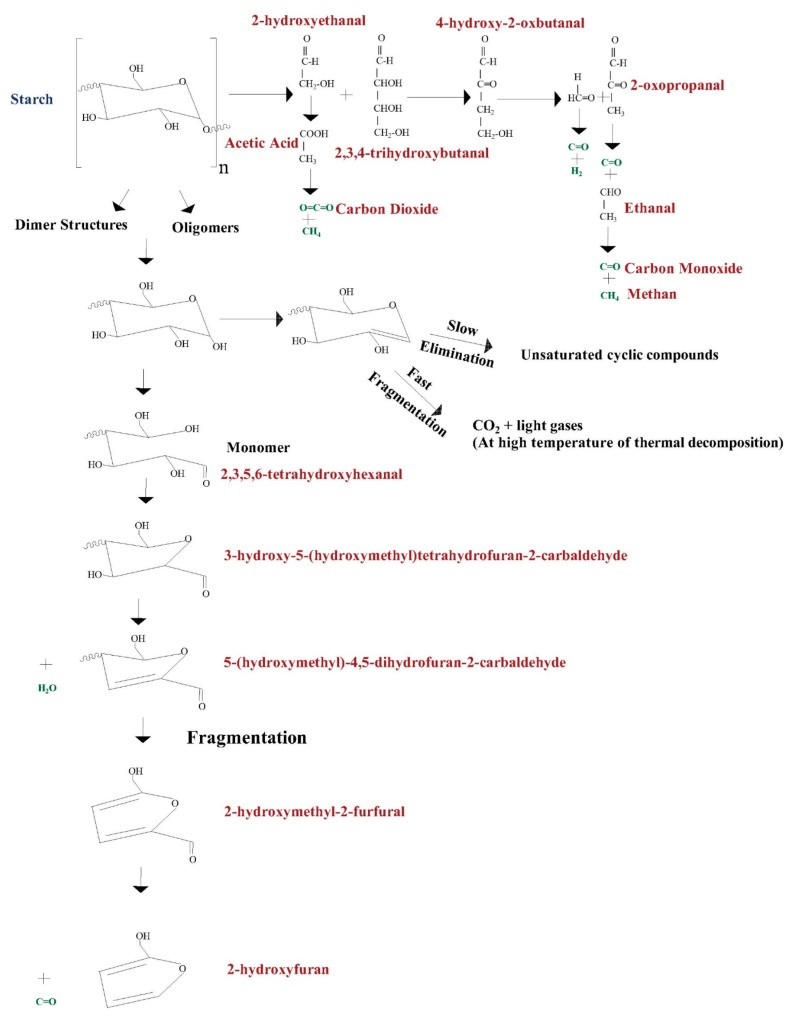
Possible pathways during starch degradation [17].

**Figure 4 polymers-12-00357-f004:**
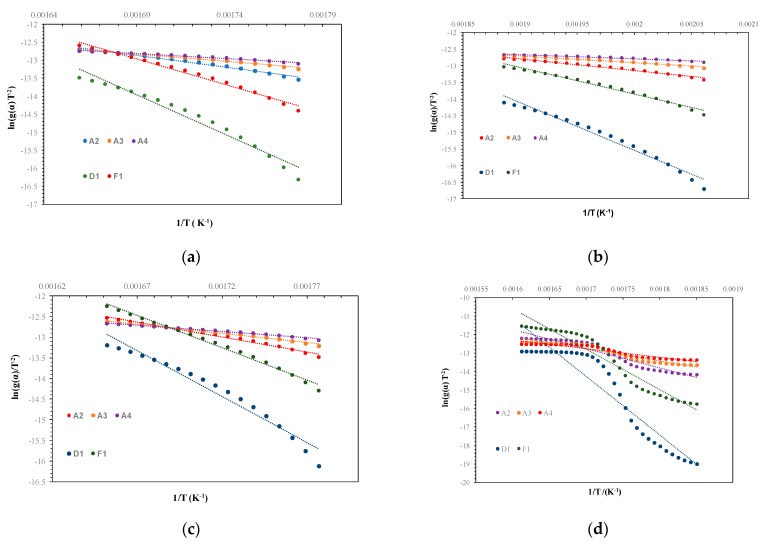
Plot ln(*g*(*α*)*T**^−^*^2^) versus 1/*T* of (**a**) rice starch; (**b**) potato starch; (**c**) wheat starch; and (**d**) corn starch.

**Figure 5 polymers-12-00357-f005:**
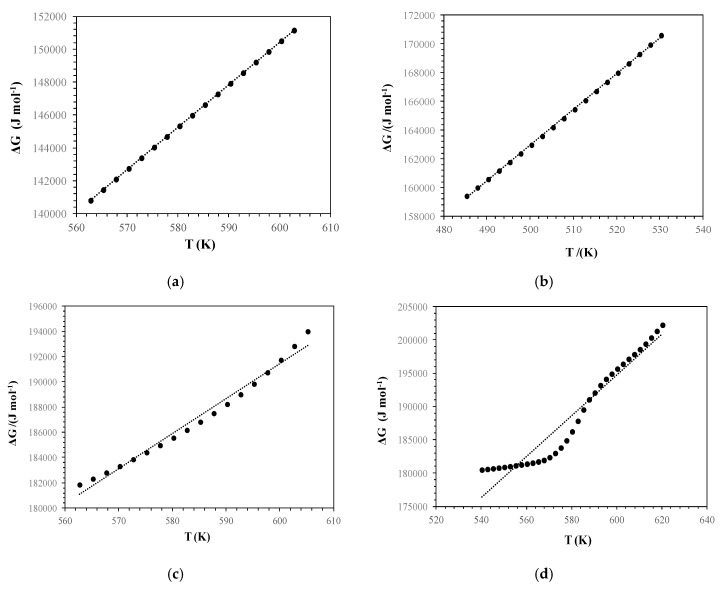
Free enthalpy versus temperature plot of (**a**) rice starch; (**b**) potato starch; (**c**) wheat starch; and (**d**) corn starch.

**Table 1 polymers-12-00357-t001:** Algebraic expressions of *g*(*α*) and *f*(*α*) for the kinetic mechanisms of the Coats–Redfern (CR) model used [15].

Mechanism	*g*(*α*)	*f*(*α*)
A_2_, Random nucleation and growth, Avrami Erofe’ve Equation	$\sqrt{(-\ln\left(1-\alpha \right)}$)	$2\left(1-\alpha \right)\;\sqrt{(-\ln\left(1-\alpha \right)})$
A_3_, Random nucleation and growth, Avrami Erofe’ve Equation	$\sqrt[{3}]{\left(-\ln\left(1-\alpha \right)\right)}$	$3\left(1-\alpha \right){\left(-\ln\left(1-\alpha \right)\right)}^{\frac{2}{3}}$
A_4_, Random nucleation and growth, Avrami Erofe’ve Equation	$\sqrt[{4}]{\left(-\ln\left(1-\alpha \right)\right)}$	$4\left(1-\alpha \right){\left(-\ln\left(1-\alpha \right)\right)}^{\frac{3}{4}}$
R_1_, Unidimensional contraction	α	1
R_2_, Cylindrical phase boundary	$\sqrt{(1-\ln\left(1-\alpha \right)}$)	$2\sqrt{\left(1-\alpha \right)}$
R_3_, Spherical phase boundary	$\sqrt[{3}]{\left(1-\ln\left(1-\alpha \right)\right)}$	$3{\left(1-\alpha \right)}^{\frac{2}{3}}$
D_1_, One-dimensional diffusion	$\alpha^{2}$	$\frac{1}{2}\alpha^{-1}$
D_2_, Two-dimensional diffusion (Valensi Equation)	(1−α)ln(1−α)+ α	${\left(-\ln\left(1-\alpha \right)\right)}^{-1}$
D_3_, Three-dimensional diffusion (Jander Equation)	${\left(1-\sqrt[{3}]{\left(1-\alpha \right)}\right)}^{2}$	$\frac{3}{2}{\left(1-{\left(1-\alpha \right)}^{\frac{1}{3}}\right)}^{-1}{\left(1-\alpha \right)}^{\frac{2}{3}}$
D_4_, Three-dimensional diffusion (Ginstling–Brounshtein Equation)	$\left(1-\left(\frac{2}{3}\right)\alpha \right)-{\left(1-\alpha \right)}^{\frac{2}{3}}$	$\frac{3}{2}{\left(1-{\left(1-\alpha \right)}^{\frac{1}{3}}\right)}^{-1}$
F_1_, First order (random nucleation with one nucleus on the individual particle)	−ln(1−α)	1−α
F_2_, Second-order (random nucleation with two nuclei on the individual particle)	${\left(1-\alpha \right)}^{-1}$	${\left(1-\alpha \right)}^{2}$
F_3_, Third-order (random nucleation with three nuclei on the individual particle)	${\left(1-\alpha \right)}^{-2}$	$\frac{1}{2}{\left(1-\alpha \right)}^{3}$

**Table 2 polymers-12-00357-t002:** Linear fit of plots for recommended adopted kinetic mechanism for the CR model.

Kinetic Model	R^2^ Rice Starch	R^2^ Potato Starch	R^2^ Wheat Starch	R^2^ Corn Starch
A_2_	0.9842	0.9771	0.9893	0.9134
A_3_	0.9809	0.9682	0.9872	0.9023
A_4_	0.9766	0.9535	0.9844	0.8892
D_1_	0.9645	0.9743	0.9599	0.8521
F_1_	0.9867	0.9828	0.9909	0.9228

**Table 3 polymers-12-00357-t003:** Estimation of the kinetic and thermodynamic parameters of starch degradation.

	Starch	RICE	POTATO	WHEAT	CORN
Parameter					
Mechanism		F_1_	F_1_	F_1_	F_1_
R^2^		0.9867	0.9828	0.9909	0.9228
Linear regression equation		y = −14890x + 12.199	y = −8003.6x + 2.1524	y = −15653x + 13.67	y = −20078x + 21.231
*E_a_* (kJ·mol^−1^)		123.733	66.542	130.139	166.928
*A* (min^−1^)		2.96 × 10^10^	6.89 × 10^5^	1.35 × 10^11^	3.34 × 10^14^
ln*A*		24.110	13.44	25.631	33.441
*k* (min^−1^)		1.301 × 10^−2^	7.115 × 10^−3^	1.387 × 10^−2^	7.995 × 10^−3^
Δ*G* (kJ·mol^−1^)		145.963	164.886	186.995	188.768
Δ*H* (kJ·mol^−1^ = *Q_p_*)		118.887	62.320	125.283	162.104
Δ*S* (kJ·mol^−1^·K^−1^)		−46.264	−201.961	−105.583	−45.504
*K* (-)		8.25 × 10^−14^	1.114 × 10^−17^	1.896 × 10^−17^	1.101 × 10^−17^

**Table 4 polymers-12-00357-t004:** Hydroxyl groups and physisorbed water determination on the surface of starches [22].

STARCH	RICE	POTATO	WHEAT	CORN
Initial mass (mg)	11.8	12.3	13.4	15.1
Final mass (mg)	1.76	2.88	2.04	2.43
Whole mass loss (%)	85.059	76.591	84.789	83.920
Whole mass loss (g)	0.01004	0.00942	0.01136	0.01267
Mass loss associated with water release (%)	9.8276	6.2299	9.8096	9.708
*n*H_2_O (mg·g^−1^ of starch) (physisorption)	98.276	62.299	98.096	97.080
*n*H_2_O (mmol·g^−1^ of starch)	5.46	3.46	5.45	5.39
Mass loss associated with hydroxyl groups release (%)	54.2176	50.8533	53.7761	57.1315
*n*OH (mol·g^−1^ of starch)	0.06019	0.05646	0.05970	0.06343
*n*OH (mmol·g^−1^ of starch)	60.24	56.46	59.70	63.43
*n*OH (mg·g^−1^ of starch)	1023.74	960.21	1015.40	1078.76

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
