# Peer review of "Kinetics and Thermodynamics of Thermal Degradation of Different Starches and Estimation the OH Group and H2O Content on the Surface by TG/DTG-DTA"

_polymers, 2020, doi:10.3390/polym12020357_

Round 1

Reviewer 1 Report

Dear Authors,

please find my comments and suggestions in attachment.

Best regards

Author Response

Thank you for your insightful review of our work, which contributed to a better understanding of scientific problems relating to the subject of the publication, and will help to eliminate potential errors in the future.Our comments and changes are noted below, and are marked in yellow in the manuscript.

Reviewer 1

Introduction: 

Question:

Why is Figure 1 is placed under the Introduction, when there is no reference to it in the introduction? The reference to this figure is before the Conclusions, almost at the very end of the manuscript.

Answer:
Thank You for that note, the reference been already placed in the introduction to this Figure.

Materials and Methods:

Question:

Materials: The description of the tested samples only shows its biological origin (and commercial source), is it possible to add more details about the tested material? What was the reason for selecting these specific samples for testing? Was the moisture content of the starch taken into account?

Answer:
We agree, it wasn’t clearly defined, because Sigma didn’t add moisture content for all samples and we only knew the pH from the Specification Sheet. So for some moisture content. To be honest, it was the main aim of this study to determine OH groups content (when it is about physicochemical properties of starches), but also the moisture was obtained for all starches and placed in Table 4 as “Mass loss associated with water release (%)”. The reason, why we decided to test those samples we justified through the most popular in the nature biological origins of starches, easy available. We were thinking about waxy starches, but, as it has been noted, we wanted to show the starches, that are easy to find in the nature. They are also really repeatable in many other studies, such as carbonization processes and they have really similar amounts of amylose and amylopectin content. So the way to modify them is much easier when we take into account the choice which one could be used in any application- just to find any difference that makes one of them better for some uses. 

We did not directly determine moisture content in our samples. Literature does not exceed 10% for such systems. The suggestion of defining it will definitely be useful for us in our research, for what we would like to thank you.

Question:

TGA experimental: What is the gas flow rate? Why this temperature range for all samples was chosen?

Answer:
Thank You so much for this note because we also confirm it should be added –  the gas flow rate was 20 mL/min. Temperature range for OH content determination was chosen for all samples based on literature. In addition, optimization of the preparation of the analysis was carried out, based on previous research results on samples with similar structure.

Question:

It was precisely written what was the mass of samples (<20 mg) taken into account in the experiment. Does this mean that the results presented in the paper are the result of only four TGA measurements (one for each starch sample)? It was assumed that the results of thermal analysis for these selected samples are always repeatable? Or maybe the measurement was always carried out on the same sample mass (i.e. 11.8, 12.3, 13.4, 15.1 mg)? And if so, why were such amounts selected? Or are representative results of measurements presented in the paper?

Answer:
They were carried out more of the measurements with different sample mass to confirm exactly the specific mass losses in different temperatures. It should be noted, that it could be circa 10-15 mg of sample mass taken (10% of error is taken into account) and the thermal analysis was repeatable always. Just in the manuscript, there were specified exact process parameters. In our measurements with different amounts of samples, we especially wanted to support the instrument’s requirements (TG/DTG-DTA). Of course, we have just clarify it in the manuscript. It should be also noted, that the most important in creating the graphs of TG (Y axis) is the percentage mass loss and graphs were almost identical. The initial mass, of course, was the reference in Table 4 just to show exactly the mechanism of our mathematical calculations and to leave no doubt if someone would like to try this calculation. Thank You so much for Your note and we hope we answered fully Your questions.

Calculations and results: 

In Figure 2b it is not clear which curve (DTG or DTA) should be analyzed according to left or right axis. Perhaps it would be more appropriate to mark e.g. the DTG curve with a dashed line? The reference to the figure should be in the text before the Line 111: "(...) R- gas constant reaching the value of 8.314 [J mol-1 K-1](...). Decimal separator: the comma should be replaced by a dot. Line 119: "where kb is the Boltzmann constant equal approximately 1.3806E-23 J K-1 (...). The comma should be replaced by a dot. Line 120: “constant equal circa 6.6261E-34 J s”. Decimal separator: the comma should be replaced by a dot.

Answer:

All comments have been corrected in the text and drawings.

Question:

There is no reference in the text to equation (12). Why?
Answer:
Thank You for that note. You are right, we have included in point 3.2 “To obtain the kinetics of starch degradation, graphs (ln (g (α) T-2)) versus (T-1) for each g (α) were created.”, but we did not refer to a specific Equation 12. It has been changed.

Comments: 

Line 109-111: units are enclosed in square brackets, however, referring to Instructions for Authors "In the text, reference numbers should be placed in square brackets [ ],...". So, this should to be corrected. In the plots, the units after the axis titles should be placed in parentheses.

Question:

In my opinion, Figure 1 is a bit infantile and overloaded with content (and graphical effects). It’s great for unconventional scientific poster, but in article it is unnecessary. The quality of the Figure in manuscript is poor.
Answer:
This figure was applied because we wanted to indicate what methods exist and show that we use one of them and it gives a lot of possible applications. We wanted to take it into account, that graphical presentation of those aspects could be clearly seen by readers. The biodegradability aspect is really important to us and very popular to many studies nowadays. We also wanted to clarify, that starch is the energy plant, which is a valuable natural biomass, so the process of biodegradation was not to overlooked by us. Thank You so much for Your suggestion. Line 133: For the graph the best fits, the slope and intercept values are used to determine Ea and A (...). Ea - "a" subscript. Compare “a" from the equations with "a" in the text ("a" in the equations is not the same letter "a" as in the text - Yes, that's the difference!) there are a lot of unnecessary double spaces in the text In the formulas, the symbols (e.g. k, h, G, H) are in italics, but not in the text and tables. Why?

Question:

Table 1: 'A3, Random nucleation and growth, Avrami Erofe've Equation', '3' - subscript

Answer:

Of course. That was really helpful for us, we have not noticed it yet. Thank You very much, it was corrected.

Question:

Figure 3: I don't know if this is a problem with this source file, but the quality of the figure is poor.

Answer:

Thank You for that suggestion. We hope the figure is more readable now.

Question:

Line 297: “α-1.6 linkages” – or ’α-1,6 linkages”?

Answer:

Thank You, that was a typo, but really important. Corrected. Thank You once again.

Question:

The same units (e.g. g, mg) are once written right behind the value (without spaces) and once separated by a space. What does that depend on?
Answer:

It should be always behind the value with spaces. Thank You. The units were corrected in the manuscript.

All comments have been corrected! Thank you very much for the thorought analysis of the work.

Reviewer 2 Report

I have been asked to review a manuscript entitled: “Kinetics of thermal degradation of different starches and estimation the OH group and H2O content on the 4 surface by TGA”.

The investigation was well conducted and authors used some techniques that allowed obtain results which were supported by literature. TG/DTG-DTA techniques were employed as well as kinetic and equations. The obtained results were well interpreted. They are relevant and interesting for the scientific community. The study can be published after minor revision, as follows:

1) Please, in page 3, line 84, explain if the used instrument is an simultaneous TG-DTA.

2) In this section, is necessary that authors quote the type of crucible used in the study (alumina, platinum, or other). They were with lid, perfurated lid or without. These are important informations in reproducibility of results.

3) In page 7, line 154-155: The sentence “TG graph for potato starch may look different, because of differences in structure caused by different amount of polymer compounds so the crystalline and amorphous region”, I think that it can be: “TG graph for potato starch may look different, because of differences in structure caused by different amount of polymer compounds so the crystalline and amorphous region as well as the granule size”.

4) My suggestion is that authors employ the abbreviation TG/DTG-DTA in all manuscript, according ICTAC nomenclature (Pure Appl. Chem. 86. (4), (2014), 545-553.

Author Response

Thank you for your insightful review of our work, which contributed to a better understanding of scientific problems relating to the subject of the publication, and will help to eliminate potential errors in the future.Our comments and changes are noted below, and are marked in yellow in the manuscript.

All comments have been corrected! Thank you very much for the thorought analysis of the work.

Reviewer 2

Question:
1) Please, in page 3, line 84, explain if the used instrument is an simultaneous TG-DTA.
Answer:
Thank You for Your note. All the analyses were carried out in TG/DTG-DTA. It has been already changed in manuscript.

Question:
2) In this section, is necessary that authors quote the type of crucible used in the study (alumina, platinum, or other). They were with lid, perfurated lid or without. These are important informations in reproducibility of results.
Answer:
Thank You for Your note. We used crucible of Al2O3 and they were without any lid. Obviously, quoting other authors regarding the crucibles used were included in the text.
Alumina crucibles are used for measurements at temperatures up to 1600 ° C. Their advantages are:
a) large volume - increases sensitivity;
b) material purity - eliminates interaction between sample and crucible;
c) material - guarantees high dimensional stability;
d) flat surface - provides excellent thermal contact.

Question:
3) In page 7, line 154-155: The sentence “TG graph for potato starch may look different, because of differences in structure caused by different amount of polymer compounds so the crystalline and amorphous region”, I think that it can be: “TG graph for potato starch may look different, because of differences in structure caused by different amount of polymer compounds so the crystalline and amorphous region as well as the granule size”.
Answer:
Thank You for Your suggestion. You are right, it sounds a lot better than the old sentence and takes into account a very important factor such as granule size. It was also corrected in the manuscript.

Question:
4) My suggestion is that authors employ the abbreviation TG/DTG-DTA in all manuscript, according ICTAC nomenclature (Pure Appl. Chem. 86. (4), (2014), 545-553.
Answer:
Thank You for Your suggestion. It has been changed in all manuscript. Only, where it was needed, so in places where we mentioned only TG or only DTA or DTG analysis, they stayed separated to highlight them from other techniques.

All comments have been corrected! Thank you very much for the thorought analysis of the work.

Thank you for the opportunity to improve and modify our work.

Beata Kurc

Corresponing Author

Reviewer 3 Report

Reserve starch is one of the most abundant polysaccharides in the world. It is essentially composed of glucosyl residues but, despite its apparently simple composition, it is structurally complex as it actually contains two types of macromolecules, amylopectin and amylose, that both are structurally divers and, depending on the plant material and the physiological state of the plant, allow complex changes.

 The manuscript focuses on thermal decomposition of four reserve starches that all are commercially available. In the present state, the manuscript is not easy to read. For instance, in the Abstract, the Coats-Redfern method is mentioned without any brief explanation (line 18).

In lines 42-44, amylose is referred to as strictly linear molecule but in reality it is a complex mixture of poorly branched and strictly linear polyglucans. Amylopectin molecules contribute both to crystalline and amorphous regions of starch as the branching points are clustered in amylopectin. Furthermore, what means 'transglicosidase' and 'glicerol' (line 63 and 78, respectively). The abbreviations TGA et al. should be briefly explained (e.g. line 83f). In Fig. 2a one graph seems to be lacking. I also do not like the term 'starch molecule'  (line 266, see also line 275f).

Author Response

Thank you for your insightful review of our work, which contributed to a better understanding of scientific problems relating to the subject of the publication, and will help to eliminate potential errors in the future.Our comments and changes are noted below, and are marked in yellow in the manuscript.

Reviewer 3

Question:
The manuscript focuses on thermal decomposition of four reserve starches that all are commercially available. In the present state, the manuscript is not easy to read. For instance, in the Abstract, the Coats-Redfern method is mentioned without any brief explanation (line 18).
Answer:
Thank You for Your note, the explanation was already given earlier in the Abstract as “This procedure is commonly used TGA kinetic method for single rate process.” in the next sentence. This method utilizes the asymptotic series expansion for approximating the exponential integral, given by Equation 11:

We plotted, as it was recommended graphs:  g(α)/T2 versus 1/T for all the mechanisms form Table 1. After chasing the model which graph was the most linear R=1, the selected mechanism was chosen) we obtained Ea and A from the slope and intercept, respectively.  This procedure is also more explained in Table 1, where are placed all kinetic mechanisms and their algebraic expressions of g(α) and f(α) for that model. There are also many other models for solid state kinetic reactions, f.e.: Broido (BR), van Krevelen (VK), but this method is often used when there is only one heating rate used. This procedure was used to determine kinetics parameters, such as activation energy, pre-exponential factor or order of nucleation process of thermal decomposition of biomass in non-isothermal TGA. All equations and explanations are based on Reference : 15.     Merci A., Mali S., Carvalho G., Waxy maize, corn and cassava starch: Thermal degradation kinetics. Ciências Exatas e Tecnológicas, Londrina, 2019, 40 (1), 13-22, doi: 10.5433/1679-0375.2019v40n1p13.

Question:
In lines 42-44, amylose is referred to as strictly linear molecule but in reality it is a complex mixture of poorly branched and strictly linear polyglucans. Amylopectin molecules contribute both to crystalline and amorphous regions of starch as the branching points are clustered in amylopectin.
Answer: Thank You for Your suggestion. To be honest, Your comment is really useful for us, because we thought the same way as You do. But after the research of literature (we wanted to confirm our knowledge). There are so many different opinions, hence the justifications. However, we agree with Your note and we have already clarified it in the manuscript.

Question:
Furthermore, what means 'transglicosidase' and 'glicerol' (line 63 and 78, respectively). Answer:
The transglucosidation process is the intermolecular bond formation, where glucosidic linkages are formed (such as 1à2; 1à3; 1à4; 1à6). Glicerol is the organic 1,2,3-Propanetriol and was used here to mark its presence in Potato Starch (modification made by Sigma). The explanations of those phrases were also placed in manuscript.

Question:
The abbreviations TGA et al. should be briefly explained (e.g. line 83f).
Answer: Thank You. It has been already changed and TGA, DTA, DTG abbreviations were explained in manuscript.

Question:
In Fig. 2a one graph seems to be lacking.

Answer:
We know what You mean. It is not lacking, because it is seen (not clearly I know), but the TG analysis for three starches (Corn, Rice, Wheat) seemed to be really similar, especially after the temperature of circa 360 Celsius Grad. So I guess that not visible one graph is Rice Starch, which is marked with blue line “under the Corn Starch”, so the yellow line. We will precise this inside the manuscript. Thank You for Your suggestion.

Question:
I also do not like the term 'starch molecule'  (line 266, see also line 275f).

Answer:
Thank You for that note. It will be replaced with ‘on the starch surface’. It seems more correctly and precisely. We agree with You.

Thank you for the opportunity to improve and modify our work.

Beata Kurc

Corresponing Author

Round 2

Reviewer 3 Report

During Revision, some improvement of the manuscript has been achieved.

Therefore, I would not oppose publication of the revised Version of the manuscript.